# Characterization of Female US Marine Recruits: Workload, Caloric Expenditure, Fitness, Injury Rates, and Menstrual Cycle Disruption during Bootcamp

**DOI:** 10.3390/nu15071639

**Published:** 2023-03-28

**Authors:** Andrea C. Givens, Jake R. Bernards, Karen R. Kelly

**Affiliations:** 1Leidos, Inc., San Diego, CA 92121, USA; 2Applied Translational Exercise and Metabolic Physiology Team, Warfighter Performance, Naval Health Research Center, San Diego, CA 92106, USA

**Keywords:** women performance, exercise, wearables, sports nutrition, RED-S, military basic training

## Abstract

Basic training is centered on developing the physical and tactical skills essential to train a recruit into a Marine. The abrupt increase in activity and energy expenditure in young recruits may contribute to high rates of musculoskeletal injuries, to which females are more susceptible. To date, the total workload of United State Marine Corps (USMC) bootcamp is unknown and should include movement around the military base (e.g., to and from dining facilities, training locations, and classrooms). Thus, the purpose of this effort was to quantify workload and caloric expenditure, as well as qualitatively assess the impact of female reproductive health and injury rates in female recruits. Female recruits (n = 79; age: 19.1 ± 0.2 years, weight: 59.6 ± 0.8 kg, height: 161.6 ± 0.7 cm) wore physiological monitors daily throughout 10 weeks of USMC bootcamp. Physical fitness test scores, physiological metrics from wearables, injury data, and menstrual cycle information were obtained. Female recruits on average expended 3096 ± 9 kcal per day, walked 11.0 ± 0.1 miles per day, and slept 5:43 ± 1:06 h:min per night throughout the 10 weeks of bootcamp. About one-third (35%) of female recruits sustained an injury. In a subset of females that were not taking birth control and had previously been menstruating, 85% experienced cycle dysfunction during boot camp. High levels of physical activity and caloric expenditure, coupled with the stress of a new environment and insufficient sleep, may lead to alterations in female reproductive cycles and musculoskeletal injuries in young USMC recruits.

## 1. Introduction

Exercise stress, reduced energy intake, and poor sleep can lead to abnormal sex-hormone production [1,2,3,4,5]. Internal and external stress elicits the same stress response through a concerted release of hormones along the hypothalamic–pituitary–adrenal axis (HPAA) and hypothalamic–pituitary–gonadal axis (HPGA), which function to maintain a balance of hormones in response to stress. Chronic exposure to stress can alter HPAA function and reproductive endocrine dysfunction of the HPGA [4]. These alterations affect both males and females, with the prevalence of data in military populations on males [6,7] and in sport on females [8]. Now, with the increased number of females entering historically male military roles, incidences of reproductive dysfunction in females during military training have been noted [9,10]. The focus to date from a military women’s health perspective has been primarily on the prevention of unintended pregnancy and suppression of menstrual cycle while deployed [11]. Recently, an observational study in British female military women during training reported a suppression of the HPGA that may have resulted from an increase in activation of HPAA [12], although stress hormones were not measured. The authors noted that a complex interplay between stressors, beyond purely physical, might be responsible for endocrine changes. However, the largest gap that has been identified is the quantification of total physical workload of basic training outside of what is accounted for in the program of instruction (POI).

United States Marine Corps (USMC) entry-level training (e.g., bootcamp) is a 13 week period of introductory training that is split into processing and administration weeks (week 0 and weeks 11–12, respectively), with 10 weeks of tactical training, operational skills, and varying physical activities that follow gender-neutral standards (weeks 1–10). While females have excelled in many military occupational roles, injury rates among females are significantly higher than males [13,14,15]. Basic training follows a POI; however, the POI is centered on the physical activities and movements essential to train a recruit into a Marine and fails to account for total movement (work) throughout the day, such as walking to and from dining facilities, training locations, and classrooms. Cumulatively, the total physical workload or time under tension can be significantly higher than expected even if at low intensity (e.g., walking) [16,17]. This elevated activity volume requires increased energy needs which may not be accounted for.

The estimated caloric expenditure during military basic training programs has been reported to be 3238–4302 kcal per day in males [17,18] and 2847–3390 kcal per day for females [2,19]. This abrupt rise in energy output combined with stress of a new environment can result in an energy mismatch if caloric intake fails to match the increase in expenditure. This physiological phenomenon, known as low energy availability (LEA), results in a preservation of resources (macronutrients) for basic life function and sustainment of physical training [20]. LEA and the associated symptoms of relative energy deficiency (RED-S) have been noted in sport [21]. RED-S is a syndrome comprising impaired bone health, immunity, cardiovascular health, and physiological dysfunction to include menstrual cycle dysregulation in females. Recently, incidences of RED-S have been documented in military personnel [1], leading to the term RED-M (military equivalent of RED-S) [22]; however, data are limited, especially for early career military personnel. Concern in young service members is risk for poor bone health, including altered microstructure and decreased bone strength that have been documented in LEA [1,23,24]. Furthermore, muscular power, volume of weight lifted, and endurance capacity are reduced with LEA. Reduced strength translates to a decrease in occupational task performance [2], as well as potential alteration in biomechanics of the movement, which may increase injury.

To address potential issues with reproductive dysfunction that may contribute to injury, it is paramount to first document workload and energy expenditure to better understand the potential cumulative stress load and potential negative physiological consequences. Thus, the purpose of this effort was to quantify total physical workload, caloric expenditure, sleep, and injuries in females during USMC bootcamp, as well as assess changes in their reproductive health.

## 2. Materials and Methods

Participants. Eighty-eight female recruits across two different training Battalions at the Marine Corps Recruit Depot in San Diego, CA, USA, volunteered to participate in the study after providing written informed consent. Recruits attended bootcamp from either February to May 2021 (*n* = 30) or October 2021 to January 2022 (*n* = 58). The study protocol (NHRC.2020.0008) was approved by the Naval Health Research Center Institutional Review Board in compliance with all applicable Federal regulations governing the protection of human subjects.

Fitness parameters. All recruits are required to undergo three different physical fitness tests throughout training: initial strength test (IST) at week 0, physical fitness test (PFT) during weeks 4 and 6, and combat fitness test (CFT) during weeks 5 and 9. For the current study, recruits performed these fitness tests as normal throughout their program of instruction. All tests were administered and scored by their drill instructors, and raw scores were provided to research staff. Individuals were required to pass the IST to proceed with recruit training, which consisted of maximum repetitions of pull-ups or push-ups in 2 min, a timed plank or maximum repetition crunches in 2 min, and a timed 1.5 mile run. USMC-established minimum standards for the IST for females are one pull-up or 15 push-ups, 1:03 min plank hold or 44 crunches, and 1.5 mile run in less than 15:00 min. The PFT, designed to evaluate stamina and physical conditioning, consisted of pull-ups or push-ups, timed crunches or plank, and a timed 3 mile run. The PFT was performed twice during recruit training during weeks 4 and 6, corresponding to training days 22 (initial PFT) and 30 (final PFT). The CFT is designed to measure functional fitness and simulates battle demands in full combat utility uniforms, consisting of three back-to-back events with a 3 min rest period in between: movement to contact (MTC), an 880-yard sprint; Ammo can lift (ACL), lift a 30 lb. ammunition can overhead until elbows lock out, with as many repetitions as possible in 2 min; maneuver under fire (MANUF), a timed 300 yd shuttle run including crawls, ammunition resupply, grenade throwing, agility running, and casualty drags. The CFT is performed twice during recruit training during weeks 5 and 9, corresponding to training days 28 (initial CFT) and 53 (final CFT).

Physiological monitoring. All recruits were provided with a Polar Grit X wrist-wearable physiological monitor (Polar Electro, Kempele, Finland) [25] to wear continuously. Watches were distributed during week 1 and were collected at the end of week 10 after the conclusion of phase 3, which ended the physical training portion of bootcamp. Weeks 11–12, “Marine week”, consisted of administration tasks and graduation which were intentionally excluded from data collection. Wearable devices were programmed with individual recruit demographic information (i.e., age, height, and weight) for native algorithms that rely on this information for prediction of outcome metrics. Watches were worn continuously and were removed only ~6 h per week when collected by research staff for charging and data download. Watches estimated caloric expenditure from heart rate, nightly sleep duration when >4 h (minimum set by Polar due to algorithm for determining sleep staging), and daily mileage estimated from step count. Data were included when daily active time was at least 4 h. Active time is defined by Polar as time spent on feet or on the move. Non-wear time was indeterminable; thus, the minimum 4 h daily active time was used to capture typical recruit schedule including “off” weekend days, as well as light-duty days. Data yield represents the percentage of recruits that had usable data (i.e., greater than 4 h active time) by week. Basal metabolic rate was calculated using the Mifflin–St Jeor equation for women: 10 × weight (kg) + 6.25 × height (cm) − 5 × age (years) − 161 [26].

Reproductive health history. To assess potential changes to menstrual cycle during bootcamp, a reproductive health history was obtained to gather information regarding age at menarche, menstrual cycle length and regularity over past 12 months, and all past and current use and methods of birth control.

Injuries. Injury data were obtained from the MCRD San Diego’s Marine Corps Training Information Management System. All recruits that were treated at the Sports Medicine Clinic were documented in the record system.

### Statistical Analysis

Descriptive statistics were calculated and are presented as the mean ± standard error. Fitness test scores between timepoints were compared using *t*-tests (SPSS version 25, IBM).

## 3. Results

### 3.1. Participants

Female recruit data were included in the results if they graduated USMC bootcamp and had all fitness test scores (IST, initial and final PFT, and initial and final CFT). In total, data from 79 female recruits are reported. Demographics are presented in Table 1.

### 3.2. Fitness Test Data

Raw scores from the IST, PFT, and CFT are presented in Table 2 and Table 3. Recruits that could not complete the minimum pull-up reps (one) performed push-ups as an alternate. A plank hold was an option as an alternate to crunches. All recruits that performed a plank hold did so for the maximum time of 4:20 min:s. The percentage of recruits that performed the standard exercise (pull-ups and crunches) is reported.

#### 3.2.1. IST

Raw IST scores appear in Table 2. Of the recruits that performed pull-ups, 42 out of 61 (69%) performed 1–6 pull-ups, and 19 out of 61 (31%) performed 7–15 pull-ups.

#### 3.2.2. PFT

Raw PFT scores appear in Table 2. From the initial PFT to the final PFT (eight training days apart), the average number of pull-ups did not increase; however, the number of recruits performing 7–15 pull-ups increased from 18 out of 47 (40%) to 25 out of 51 (49%). The average number of push-ups increased by six reps, from 44 ± 2 during the initial PFT to 50 ± 2 during the final PFT (*p* < 0.02). The average number of crunches also increased by 13 reps, from 88 ± 2 during the initial PFT to 101 ± 3 during the final PFT (*p* < 0.01). Run times did not significantly improve.

#### 3.2.3. CFT

Raw CFT scores appear in Table 3. Ammo can lifts improved, increasing by 13 reps, from 66 ± 2 during the initial CFT to 79 ± 2 during the final CFT (*p* < 0.01). Maneuver under fire times improved by 8 s, from 3:04 ± 0:02 during the initial CFT to 2:56 ± 0:02 during the final CFT (*p* < 0.01). Movement to contact times did not improve from the initial CFT to the final CFT (*p* = 0.10).

### 3.3. Estimated Caloric Expenditure

The average daily caloric expenditure was 3096 ± 9 kcal/day, ranging from 1147 to 5790 kcal/day. Daily averages by training week appear in Table 4 and Figure 1. Caloric expenditure (kcal/day) was only included if the watch recorded at least 4 h or more of active time per day. Active time is defined by Polar as time spent on feet or on the move. Non-wear time was indeterminable; thus, the minimum 4 h daily active time was chosen to capture typical recruit schedule including “off” weekend days, as well as light-duty days. The highest caloric expenditure period was the final week of training, which included a 54 h culminating event, “the crucible,” where recruits completed eight major training events: a day movement resupply, a combat assault course, a casualty evacuation, a reaction course, an unknown distance firing course, a night infiltration course, and a night march. Daily caloric expenditure during the 54 h crucible was 3572 ± 48 kcal per day overall, with the final day caloric expenditure averaging 3850 ± 88 kcal, up to a maximum of 5387 kcal.

### 3.4. Sleep

Recruits slept on average 5:43 ± 1:01 h:min per night (Figure 1) and only slept more than 7 h [27] per night on 7 ± 1 nights of bootcamp. The shortest nights of sleep duration were the final two training weeks, “field week” (week 9) and “the crucible” (week 10), when no recruit slept more than 7 h per night. During the entirety of bootcamp, there were two recruits that never slept more than 7 h per night. All other recruits had at least one night of sleep that was greater than 7 h. Recruits that stopped menstruating slept on average 5:39 ± 0:01 h:min per night, compared to all others that slept 5:46 ± 0:01 h:min per night (*p* < 0.001).

### 3.5. Workload

Daily workload was on average 11.0 ± 0.1 miles per day. Daily averages by training week appear in Table 4 and Figure 1. Workload (miles/day) was only included if the watch recorded at least 4 h or more of active time per day. Active time is defined by Polar as time spent on feet or on the move. Non-wear time was indeterminable; thus, the minimum 4 h daily active time was chosen to capture typical recruit schedule including “off” weekend days, as well as light-duty days. The highest volume of workload occurred the last week of training, “the crucible”, when recruits averaged 16.2 ± 0.1 miles per day and logged up to 30 miles in 24 h (Figure 1).

### 3.6. Menstrual Cycle Function

Fifty-four recruits completed menstrual cycle history. Fifteen were excluded from the results because they were using hormonal birth control. Thus, the results include responses from 39 recruits that had been regularly menstruating prior to bootcamp and were not taking birth control prior to or during bootcamp. Average age at menarche was 13 ± 0.4 years. Most female recruits, 33 out of 39 (85%), stopped menstruating during bootcamp.

### 3.7. Injuries

Overall, 35% of recruits sustained an injury during boot camp, and 16% of recruits sustained multiple injuries. Most injuries (56%) were categorized as new overuse (Table 5 and Table 6). The top injury types were strains (23%), sprains (19%), pain (14%), and medial tibial stress syndrome, i.e., “shin splints” (9%). One displaced fracture and one nondisplaced stress fracture each accounted for 2% of injuries. Of the recruits that indicated their period stopped during the menstrual cycle survey, nine (27%) sustained injuries.

## 4. Discussion

To our knowledge, this is the first effort to quantify workload and fitness, estimate caloric expenditure and sleep duration, and qualitatively assess menstrual cycle history in female USMC recruits throughout 10 weeks of boot camp. Recruits, on average, expended 3096 ± 9 kcal and walked 11.0 ± 0.1 miles per day while sleeping an average of 5:43 ± 1:06 h:min per night. Furthermore, our estimates indicate that 7 h of sleep was achieved on only 7 nights over the course of 10 weeks. Thirty-five percent of female recruits sustained an injury during bootcamp, with new overuse injuries being the primary culprit. Fitness scores indicted that the females in this effort had “good” (65th percentile for age) cardiorespiratory fitness upon entry based on an estimated aerobic capacity of 42.7 ± 0.4 mL/kg/min calculated from their 1.5 mile run time [7].

Program-induced cumulative overload (PICO) [28] is a concept derived from observations in military programs that have heavy physical training and result in adverse outcomes such as injuries. This concept is focused on raising awareness to the cumulative load not only over a day but over a training program, which unintentionally overloads military personnel through failure to recognize the physical impact of military training in combination with traditional physical training practices. The POI for the current effort calls for the recruits to run a cumulative 29 miles over the course of 10 weeks [29]. In this effort, total workload based on step count was estimated to be 11.0 ± 0.1 miles per day. Location of barracks on the depot in relation to classrooms, dining hall, and training sites increased daily movements above what is accounted for in the POI, increasing time under tension for recruits. Moreover, this workload is greater than other basic training reports of 6.5–8.0 miles per day [17,18] but is consistent in that most of the work performed is at a low intensity. Despite low-intensity activity, 35% of female recruits sustained an injury. Most injuries (56%) were categorized as new overuse, due to strains and sprains, followed by acute traumatic (32%), and preexisting overuse (11%). This aligns with data published on males from the same recruit depot location [30] and those reported by others [31], suggesting that PICO may be a factor.

Injury risk is multifactorial and includes fitness level, body composition, physical demand, and volume of training. The population of females in this study was considered average compared to other military recruits [32,33], which may have contributed to injuries, as fitness and injury risk are tightly coupled [6,14,15,34,35,36]. Another contributing factor is the difference in absolute versus relative load. While not a focus of this effort, military training and the POI are gender-neutral. Males and females are required to carry the same absolute loads as indicated by the occupational demands. However, over time, this increased relative load on smaller-stature individuals, such as females, requires the individual to do more work. For example, conditioning hike pack weights start at 22 lbs., which is 17% of female recruits’ average bodyweight, increasing up to 66 lbs. on the final hike, roughly 50% bodyweight. Comparatively, the average male recruit weighed 165 lbs., making pack weight 13% to 40% of their bodyweight. Thus, smaller-stature individuals (females) are doing relatively more work over the same given time compared to their larger (male) counterparts. Thus, the increased overall load on their body may contribute to increased stress systemically [37].

Chronic stress increases allostatic load and can lead to menstrual cycle dysfunction [1,4,5]. Over the course of the 10 weeks, 85% of the female recruits stopped menstruating and had previously reported menstruating prior to onset of bootcamp. Menstrual cycle dysfunction has been well described in sport and is most related to LEA or RED-S [4,8] from inadequate calorie intake. Recruits in the present study theoretically had access to adequate calories; thus, the increased physical and psychological stress load of bootcamp may have contributed to reproductive system dysfunction. Furthermore, we did not measure any significant changes in weight over the 10 weeks, lending support to increased physical activity and cumulative stress potentially disrupting menstrual cycle. It has been reported that as many as 50% of exercising women experience menstrual disturbances [5]; however, limited data exist in military women, especially in entry level training. The sparse data available in female service members focuses on intentional cycle suppression for deployment [11]. A 2003 study conducted in military academy cadets reported that 114 out of 116 (98%) had menstrual irregularity during their freshman year [9], which aligns with the data presented herein. More recently, a 2017 study in Korean military women reported that 28 out of 40 (70%) developed irregular cycles during the training course [10]. Chronic anovulation, associated with stress, weight loss, excessive exercise, or a combination thereof, can result in serious medical complications, such as bone loss, that ultimately affect health and readiness; thus, there is a need to further understand the cause of cycle disruption in order to provide mitigation strategies. These data, while collected via self-report, are the first to our knowledge to be coupled to caloric expenditure and workload in USMC recruits. While more work is needed to elucidate the cause of menstrual cycle dysfunction, these data bring awareness to a musculoskeletal injury risk factor.

Another risk factor for injury is inadequate nutrition. The estimated caloric expenditure in the current effort is in concurrence with data previously published in female British military recruits during basic training. In that effort, double-labeled water was used over a 10 day period, and caloric expenditure was 3057 kcal per day [3], which is in line with estimations from heart rate and work load in this effort. Moreover, variability in caloric expenditure by training week was similar to the caloric expenditure in the current study ranging from approximately 2800 to 3400 kcal [2]. In the present effort, caloric expenditure was highest during the initial and final weeks of training. High expenditure in initial weeks was attributed to stress of a new environment and adjustment to workload [38], whereas high expenditure in the final weeks was due to the greatest training volume. While our present study did not measure dietary intake, mess hall meals are standardized to offer 3700 calorie menus daily based on the Military Dietary Reference Intake (MDRI), “heavy activity” level, under Marine Corps Order 10,110.49 [39]. In addition to mess hall meals, recruits are provided supplemental nutrition snacks that offer 500–600 calories during most training days; consumption is recommended but not mandatory. Thus, in theory, recruits should be in equal or positive energy balance. However, others have reported that consumption of meals may be hindered by physical and verbal interference from drill instructors, as well as time restriction [40]. Thus, while adequate nutrition is offered, energy balance might not be achieved.

There are several limitations that need to be addressed. The primary goals were to determine total workload, estimate caloric expenditure, and document injury rates in female recruits. It is acknowledged that wearables estimate caloric expenditure on the basis of heart rate; however, Polar heart rate monitors have been validated in multiple efforts [41,42,43], and the use of wearable technology in field data collection has become widely accepted [44,45,46,47]. Additionally, menstrual cycle history was collected as part of basic health history but was not an initial focus. As such, menstrual data were self-reported via recall at the end of bootcamp. Nevertheless, this information is critical to raise awareness on menstrual dysfunction in young female recruits. Furthermore, we were not able to measure dietary intake to calculate energy balance. At the onset of the study, COVID-19 restrictions created limitations on personnel that were allowed on base and indoors and in proximity of recruits. As such, research personnel were authorized to collect data outdoors which limited ability to measure caloric intake. This also impacted body composition metrics; thus, only body weight was collected. Despite these limitations, the data contained herein highlight the importance of quantifying caloric intake, as well as the need for more research relevant to “RED-M”, which is the military equivalent to RED-S [22]. Furthermore, this effort raises awareness on the potential impact of heavy physical training on reproductive health in young female military members, analogous to what has been reported in sport.

## 5. Conclusions

Bootcamp is an intense, albeit short period of high daily workload and caloric expenditure coupled with limited sleep. While recruits can perform to fitness standards, substantial injury rates and menstrual cycle dysregulation can occur [1], as reported by others and suggested in this effort, giving credence to the concept of program-induced overuse injuries. The goal of bootcamp or entry-level training is to teach an individual technical and tactical skills and maintain fitness standards; however, balancing total workload required to achieve this goal with a holistic approach of adequate nutrition, rest, and recovery should be considered. Collectively, these findings demonstrate the requirement for further work to elucidate whether cessation of menstrual cycle is related to altered hormone secretion as a function of reduced energy intake, increased energy expenditure, or a combination of both.

## Figures and Tables

**Figure 1 nutrients-15-01639-f001:**
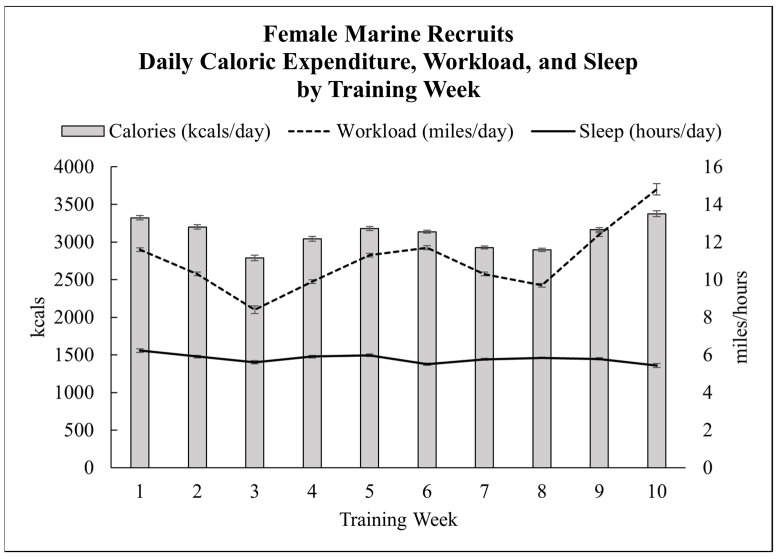
Daily workload, caloric expenditure, and sleep duration by training week.

**Table 1 nutrients-15-01639-t001:** Recruit Demographics, *n* = 79.

Age (years)	Height (cm)	Weight (kg)	BMI (kg/m^2^)	BMR (kcal)	VO_2max_ (mL/kg/min)
19.1 ± 0.2	161.6 ± 0.7	59.6 ± 0.8	22.8 ± 0.2	1349 ± 11	42.7 ± 0.4

BMI: body mass index, BMR: basal metabolic rate, estimated using Mifflin–St Jeor equation where BMR = 10 × weight (kg) + 6.25 × height (cm) − 5 × age (years) − 161 [26], VO_2max_: maximal oxygen consumption, estimated using ACSM equation where VO_2max_ (mL/kg/min) = 3.5 + 483/1.5 mile time in min [7].

**Table 2 nutrients-15-01639-t002:** Initial strength test and physical fitness test raw scores.

	IST	Initial PFT	Final PFT
	Week 0	Week 4	Week 6
Pull-ups (reps)	5 ± 1	6 ± 1	6 ± 1
Performed pull-ups *	77%	63%	67%
Push-ups (reps)	34 ± 2	44 ± 2	50 ± 2
Crunches(reps)	86 ± 2	88 ± 2	101 ± 3
Performed crunches *	100%	95%	86%
1.5 mile run (min:s)	12:24 ± 0:11	n/a	n/a
1.5 mile run (pace)	8:16	n/a	n/a
3 mile run (min:s)	n/a	25:14 ± 0:15	24:51 ± 0:15
3 mile run (pace)	n/a	8:25	8:17

Data are presented as the mean ± standard error. Pull-ups, push-ups, and crunches are the total number of repetitions (reps). * Percentage of recruits that performed standard pull-ups or crunches. Push-ups were the alternate to pull-ups. A plank hold was the alternate to crunches. All recruits that opted for the alternate plank performed the hold for the maximum time of 4:20 min:s.

**Table 3 nutrients-15-01639-t003:** Combat fitness test raw scores.

	Initial CFT	Final CFT
	Week 5	Week 9
Movement to contact (min:s)	3:31 ± 0:02	3:27 ± 0:02
Ammo can lift (reps)	66 ± 2	79 ± 2
Maneuver under fire (min:s)	3:04 ± 0:02	2:56 ± 0:02

Data are presented as the mean ± standard error. Movement to contact and maneuver under fire times are in min:s. Ammo can lift scores are the total number of repetitions (reps).

**Table 4 nutrients-15-01639-t004:** Daily caloric expenditure and workload by training week.

	Caloric Expenditure (kcal/day)		Workload (miles/day)		Data Yield
Training Week	Mean ± SE	Range	Mean ± SE	Range	
1	3324 ± 30	1503–4916	11.6 ± 0.1	4.7–20.4	98%
2	3201 ± 30	1727–5245	10.3 ± 0.1	2.6–23.4	93%
3	2789 ± 36	1437–4870	8.4 ± 0.2	2.3–16.4	73%
4	3044 ± 31	1688–5070	9.9 ± 0.1	2.7–19.8	93%
5	3178 ± 28	1699–5449	11.3 ± 0.1	2.5–24.7	96%
6	3137 ± 20	1855–4690	11.7 ± 0.1	3.8–19.8	98%
7	2927 ± 20	1147–4578	10.3 ± 0.1	3.3–18.9	99%
8	2895 ± 21	1750–4960	9.7 ± 0.1	3.3–19.7	93%
9	3165 ± 24	1678–5387	12.4 ± 0.1	3.7–21.4	92%
10	3377 ± 41	1165–5790	14.8 ± 0.3	3.5–29.5	89%

Data are presented as the mean ± standard error. Data yield is the percentage of recruits that had useable data.

**Table 5 nutrients-15-01639-t005:** Injury frequencies.

Injury Frequencies	Count	%
Total number of injuries	57	n/a
Number of recruits injured	28	35%
Number of recruits with multiple injuries	13	16%

**Table 6 nutrients-15-01639-t006:** Injury categories.

Injury Category	Count	%
New Overuse	32	56%
Acute/Traumatic	18	32%
Preexisting Overuse	6	11%

## Data Availability

The data presented in this study are available on request from the corresponding author.

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
