# Peer review of "Characterization of Female US Marine Recruits: Workload, Caloric Expenditure, Fitness, Injury Rates, and Menstrual Cycle Disruption during Bootcamp"

_nutrients, 2023, doi:10.3390/nu15071639_

Round 1
Reviewer 1 Report
The article titled „Characterization of female US Marine recruits: workload, caloric expenditure, fitness, injury rates and menstrual cycle disruption during bootcamp” by Andrea Givens et. al. is very interesting and fits to the scope of the Nutrients journal.
In this original work, the authors undertook quantify total physical workload, caloric expenditure, sleep, and injuries in females during bootcamp, as well as assess changes of their reproductive health.
The article contains all the elements, the individual sections are well developed.
There are some comments that could help to improve interesting manuscript:
In the text:
1. Minor spell check required.
2. Reference numbers should be placed in square brackets [ ], and placed before the punctuation; for example [1], [1–3] or [1,3].
3. All Figures, Schemes and Tables should be inserted into the main text close to their first citation and must be numbered following their number of appearance (Figure 1, Scheme I, Figure 2, Scheme II, Table 1, etc.).
4. References
Complete all the references in accordance with the Instructions for the authors, should be described as follows:
Journal Articles:
Author 1, A.B.; Author 2, C.D. Title of the article. Abbreviated Journal Name Year, Volume, page range.
Books and Book Chapters:
Author 1, A.; Author 2, B. Book Title, 3rd ed.; Publisher: Publisher Location, Country, Year; pp. 154–196.
Reviewer 2 Report
The introduction and background set up a good case for the study, however the study design could be better.
Measurement of dietary energy intake us necessary to determine if low energy availability exists. This manuscript has been submitted to the sports nutrition special, but the current study doesn't investigate the relationship between energy balance and the menstrual cycle.
Wrist wearables produce a lot of error when assessing energy expenditure across different modes of activity. The energy expenditure data may not accurately represent the actual EE.
The methods used to collect the data are not strong reliable assessment tools.
It would be interesting to see some stronger statistics. Comparison of those who stopped menstruating and energy, injury, sleep, etc. A paired t-test is not a strong test of statistical assessment.
Line 308 should read RED-S, not RED-M
Of the 85% who stopped menstruating during bootcamp, how many also experienced changes in energy intake?
Where are the tables and figures?
I think this is a novel study and will contribute to our understanding of how basic training can impact female health, but the current study doesn't investigate this relationship enough to draw these conclusions. Better measurement tools of energy intake, menstrual function (ie. hormones), energy expenditure, energy availability, injury rate, etc is important to understand the relationship. The current study provides descriptive information, but the conclusions drawn are extrapolated.
Author Response
We would like to thank the reviewers for their comments on this submission. We appreciate the feed back and have made edits accordingly and where appropriate. Please see the point by point response to each reviewers comments:
Response to reviewer #2:
Reviewer 2, we appreciate your feedback and we recognize these limitations you have raised and noted them ourselves which resulted in another funded grant. The limitations you raised are the thrust of our new effort starting in May 2023 and continuing until September 2025. However, in order to receive funding and to address the issue in boot-camp we first have to document the “problem” in order to obtain support and funding for future research on this topic. I have been following my British colleagues lead on raising awareness for our young female military members in order to lobby for resources to adequate study the etiology behind our initial findings. I have addressed your comments and concerns the best we can given our current data set and have added to the limitations section. We hope you find these satisfactory and find value and merit in this effort as the foundation for our current/future work.
The introduction and background set up a good case for the study, however the study design could be better.
Response: We recognize this limitation. The initial premise of our study was to quantify workload in boot camp (publication in review) and through our menstrual cycle history survey and in discussions with the females in our effort this finding was surprising. As such, we felt this finding was important for health and longevity of these young women and ironically, menstrual cycle dysfunction has never been quantified in USMC female recruits. Thus, despite the limitations, we feel that it is important to raise awareness of this issue in order for resources to be allocated to the young women that are striving to be Marines.
Measurement of dietary energy intake us necessary to determine if low energy availability exists. This manuscript has been submitted to the sports nutrition special, but the current study doesn't investigate the relationship between energy balance and the menstrual cycle.
We absolutely agree. However, this is a tremendous undertaking and requires significant disruption to a very tight minute by minute training schedule. However, based upon the first success of our effort, a plate waste study has been authorized and will commence with the next iteration of this study. The initial findings presented in this manuscript were critical in obtaining support from military leadership. The upcoming effort will address energy imbalance.
However, per Marine Corps instruction students are given 3700kcal/day to consume. It is up to the individual recruit to decide how much to eat. While energy intake was not possible during this initial effort, we do know that the recruits had access to at least 3700 kcals/day.
Wrist wearables produce a lot of error when assessing energy expenditure across different modes of activity. The energy expenditure data may not accurately represent the actual EE.
We have validated the polar grit against indirect calorimetry (paper in public affairs review). While those findings have not undergone peer review we did find that the caloric expenditure to indirect calorimetry (Parvo Medics) had a ICC of 0.854; 95% CI of 0.711-0.929, p<0.001. This was conducted across 5 different exercise intensities ranging from low to maximal exercise. For field research especially in larger cohorts and for the length of this effort 10 weeks, we need to rely on wearable technology. It was not practical to use double label water for multiple reasons to include duration of effort, access to recruits to collect samples due to COVID 19 restrictions and training locations weeks 6-10. Further, the recruits wore their monitors 24 hours a day and while there is likely a small percentage of error, this is acceptable in the context of this effort. If this were a weight loss study then I would understand the concern with estimating caloric expenditure via wearables. Wearable technology for estimates of caloric expenditure have been used and accepted for free living and field research.
We have added appropriate references and information to the limitations section to address this concern.
The methods used to collect the data are not strong reliable assessment tools.
We appreciate this comment. However as stated previously and in the manuscript this finding was to raise awareness on the issue. In the upcoming effort, we will be collecting morning void urine samples every day for 10-weeks to measure reproductive hormones as well as other reliable measures.
It would be interesting to see some stronger statistics. Comparison of those who stopped menstruating and energy, injury, sleep, etc. A paired t-test is not a strong test of statistical assessment.
Thank you for this comment. We have added some information to the results and discussion section. For sleep, there were statistically significant differences in those who stopped menstruating; however, it is clinical irrelevant as the difference was only 7 minutes. With respect to injuries, it is difficult to make meaningful conclusions. The average time an individual was on light duty was 3 days and the majority of injuries were mild in nature and either coded as sprain or pain. Out of the 58 total injuries, there were only 2 stress fractures that were acute and traumatic in nature from falling. In our upcoming effort that is going to focus on the etiology behind observed menstrual cycle dysfunction we will have a larger sample size and expect to address this comment.
Line 308 should read RED-S, not RED-M
This has been clarified.
Of the 85% who stopped menstruating during bootcamp, how many also experienced changes in energy intake?
As noted above and in the manuscript, this was not measured. It will be in our upcoming effort.
Where are the tables and figures?
We apologize but they were uploaded with the manuscript and should have been in the final pdf. This oversight will be corrected.
I think this is a novel study and will contribute to our understanding of how basic training can impact female health, but the current study doesn't investigate this relationship enough to draw these conclusions. Better measurement tools of energy intake, menstrual function (ie. hormones), energy expenditure, energy availability, injury rate, etc is important to understand the relationship. The current study provides descriptive information, but the conclusions drawn are extrapolated.
We acknowledge the limitations and appreciate the concern. We do feel this information is valuable and critical for raising awareness of this issue amongst our young female military members. With the over-emphasis on equality, we hope that this paper as well as our upcoming work will steer efforts towards increasing equity in our female military members and providing necessary resources.
Round 2
Reviewer 2 Report
Typo line 334: It is acknowledged that wearable only estimate caloric expenditure based upon heart; however Polar heart rate has been validated in multiple efforts [40-42] and the use of wearable technology in field data collection has become widely accepted.
*this is a recent edit, and had two typos in it. The edits seem rushed and sloppy.
The authors address the limitations on menstrual cycle tracking, but since this is a large focus of the study it still seems that the conclusions involving the menstrual cycle are extrapolated.
Same with energy intake data- the foundation of the study is on energy intake/energy expenditure, and the data collected are loose estimations due to poor measurement tools. Thus the strength of the results and the validity of the study is questionable.
Line 348: "the importance of quantifying caloric intake and as well as the need for more research relevant to “RED-M” which is the military equivalent to RED-S." Since RED-M is not a term widely used, it need to be better defined. Or, just use REDs, but also define what this is. Stating RED-M doesn't give the reader any idea what the authors are discussing.
Line 349: "Further, this effort raised awareness on the potential impact of heavy physical training on reproductive health in young female military members." I'm not sure you can state this based on the methods and data collected.
